# The Role of Digital Gaming in Addressing Loneliness Among Older Adults: A Scoping Review

**DOI:** 10.3390/healthcare13172140

**Published:** 2025-08-27

**Authors:** Eunie (Yoon Kyung) Jung, Jiadong Yu

**Affiliations:** 1Department of Counseling and School Psychology, Benerd College, University of the Pacific, Stockton, CA 95211, USA; 2California School of Professional Psychology, Alliant University, Fresno Campus, Fresno, CA 93727, USA

**Keywords:** older adults, digital gaming, scoping review, loneliness, social isolation

## Abstract

As the world’s population of older adults continues to grow rapidly, combating loneliness and social isolation has become an emerging health concern for this group. Though there has been increasing interest within the scientific community in exploring digital games as an intervention for loneliness, research on gaming as an intervention is a relatively new field of study. This scoping review examines the current state of research on the impact of digital gaming on loneliness in the older adult population and was conducted in accordance with the 2018 Preferred Reporting Items for Systematic Reviews and Meta-Analyses extension for Scoping Reviews (PRISMA-ScR) guidelines. A total of 317 potentially relevant studies were identified through database searches, and of these, 278 studies were excluded due to failure to meet inclusion criteria. The full texts of 39 articles were assessed for eligibility, resulting in 9 articles being included in this scoping review. Some important findings from our study include the central role of social interaction in addressing loneliness and the potential for interventions targeting both physical and mental well-being to have a more significant impact on alleviating loneliness. We also found that while many studies reported positive effects of gaming interventions, some findings were contradictory, suggesting that the relationship between gaming and loneliness is complex and moderated by multiple factors. Recommendations for future research include expanding investigations to outside of East Asia (where the majority of existing studies were conducted) to the United States, Africa, India, or Europe.

## 1. Introduction

Worldwide, the human population is growing older and living longer. The group of individuals above the age of 65 years is growing more rapidly than the demographic below that age [1]. It is projected that by 2050, the number of older persons is expected to double, reaching nearly 2.1 billion [1]. While medical advancements have extended life expectancy, aging populations face increasing social and psychological challenges, including social isolation and loneliness [2].

Loneliness, defined as the distressing experience arising from a perceived gap between desired and actual social relationships [3], is a pressing public health concern among older adults [4]. Studies indicate that nearly one in four older adults experiences chronic loneliness, with higher prevalence among those who live alone, are widowed, or experience physical impairments [5,6]. There are many challenges that accompany supporting a historically large demographic of older adults. Mental health challenges such as depression are prevalent in this group and have become a widespread global disability [7,8,9]. Many studies have examined the development of depression in the older adult population and have found that loneliness is linked to an increased risk of depression [10,11], cognitive decline [12,13], cardiovascular diseases [14], and all-cause mortality [4]. Loneliness has not only been identified as a risk factor for depression, but a large body of research shows that loneliness, with or without depressive symptoms, has a serious impact on physical and mental health, quality of life, and longevity [15,16,17]. The effect of loneliness on mortality is comparable to that of other well-established risk factors such as smoking, obesity, and physical inactivity [18]. Sounding the alarm on these findings, the Surgeon General’s Advisory recently referred to “our epidemic of loneliness and isolation” [18], a trend which is prevalent in older adult populations [5,6]. Addressing loneliness has thus become a critical priority in health and psychological research, necessitating the identification of effective and scalable interventions.

Various treatments have been studied and proposed to help alleviate loneliness in the older adult population, including behavioral activation [19], group and individual interventions [20], and evidence-based psychotherapy treatments [21]. While these approaches show promise, they are not without limitations. Many require in-person participation, which can be inaccessible due to mobility issues, financial constraints, or geographic barriers [22]. Others fail to sustain engagement over time, leading to diminishing effects [23,24]. Furthermore, interventions that rely on passive social exposure (e.g., social prescribing and telephone support groups) may not address the emotional depth needed to alleviate loneliness [25].

Additionally, the opportunity cost model of subjective effort [26] suggests that people constantly evaluate how to allocate their mental energy across competing tasks, choosing those that yield the most subjective reward. Digital games can be appealing because they offer a high perceived return for minimal effort, especially among those who are cognitively or emotionally depleted. Likewise, the Conservation of Resources (COR) theory [27] emphasizes individuals’ motivation to preserve and protect their remaining resources. Older adults who are experiencing physical or social loss may turn to low-effort, high-engagement activities like gaming to avoid further depletion of emotional or social resources, while attempting to replenish them through enjoyment or mental stimulation.

A recent meta-analysis found that digital technology interventions (DTIs) did not have significant effects on loneliness in an older adult population [28]. However, this study focused on a broad set of interventions, including internet-based social activities, videoconferencing, and video or voice networks, and did not include digital gaming. Digital gaming has been theorized to mitigate loneliness through multiple mechanisms. Multiplayer and cooperative games foster meaningful social interactions by connecting older adults with peers, family members, and intergenerational players [29]. Several studies have supported the importance of social interaction for healthy aging [30,31]. Additionally, gaming can enhance emotional well-being by increasing engagement, enjoyment, and self-efficacy, counteracting depressive symptoms linked to loneliness [32]. Certain game formats, such as exergames and puzzle-based games, provide cognitive stimulation, which may indirectly alleviate loneliness by promoting mental agility and social participation [33]. This type of cognitive stimulation has also been theorized to contribute to active and healthy aging [34].

Contrary to traditional perceptions that video games primarily appeal to younger demographics, an increasing number of older adults engage in digital gaming. Available research suggests that video game–based technology can create opportunities for social connection and help alleviate social isolation and loneliness in this age group. A growing percentage of the older adult population worldwide engages in gaming on a regular basis [35]. For example, in a 2020 survey conducted in the U.S., older adults were found to be more likely than younger adults to play games daily or weekly [36]. Though there has been increasing interest among the scientific community to explore digital games as an intervention, research on gaming as an intervention is a relatively new field of study [37]. As a complement to traditional interventions, some research suggest that these digital games can help older adults boost their emotional health by widening their social support networks [38]. However, other studies suggest that the impact of gaming on social isolation or loneliness may be minimal, and that further research needs to be conducted before coming to a more definitive conclusion about its impact on social connection [39]. For example, problematic game use has been found to be positively correlated with loneliness [40,41], suggesting that gaming may, in some cases, exacerbate rather than alleviate social isolation. Related research has also found that motives for gaming seem to correlate with both increased and decreased loneliness, suggesting a complex relationship between the reasons for gaming and their relationship to social isolation [42,43].

The purpose of this paper is to provide a scoping review of the literature on the impact of digital gaming on loneliness and social isolation in older adults. “Digital games” are defined as any game played on an electronic device, either online or independently, for example, on a computer, a video game console, a mobile device, or interactive television [44]. Our aim is to provide a current view of this developing field of research and to provide more clarity on previously reported findings that seem contradictory in nature. Indeed, researchers in the field have asserted that more studies are needed to evaluate the effectiveness of new technologies in addressing conditions including loneliness [45]. In this review, we examine the impact of video games on depression and loneliness in the older adult population through three main research questions: (1) What types of digital games are being used to help address loneliness in older adults? (2) How are studies measuring the effect of these digital games on loneliness in older adults? (3) What outcomes have been reported from these studies?

## 2. Methods and Materials

This scoping review was conducted and reported in accordance with the 2018 Preferred Reporting Items for Systematic Reviews and Meta-Analyses extension for Scoping Reviews (PRISMA-ScR) guidelines [46]. The protocol for this review also adheres to the PRISMA reporting guidelines.

### 2.1. Literature Search

To identify potentially relevant studies, we conducted a comprehensive search on 23 October 2024 across the following databases: EBSCOhost (including Communication & Mass Media Complete, Academic Search Complete, MEDLINE, CINAHL, and APA PsycInfo). Only articles published in English were considered. The search strategy employed the following keywords: (older adults OR older people OR elderly OR geriatric OR aging OR senior OR seniors) AND (loneliness OR social isolation OR social exclusion OR lonely OR social interaction) AND (gaming OR video games OR online gaming OR internet games OR online games). In addition to these databases, we conducted supplementary searches on Google Scholar using the terms “older adults”, “loneliness”, and “gaming” to capture any additional relevant studies.

### 2.2. Study Selection and Data Extraction

Data extraction was conducted independently by two reviewers and cross-checked to ensure accuracy and consistency using a data collection form that was predefined by mutual agreement. Each article was meticulously reviewed to determine eligibility based on the inclusion and exclusion criteria. The inclusion criteria were as follows: (a) the study included participants aged 65 years or older or reported a mean participant age of 65 years or above; (b) the article examined video game technology; (c) the study assessed participant loneliness using a validated measure; (d) the article was accessible through the specified databases; and (e) the study was published from 2014 to the present. The year 2014 was chosen as a starting point for this review, as research supports that digital gaming among older adults became increasingly popular around this period of time [36,47].

Articles that did not report original results (e.g., commentaries, study protocols, and letters to the editor) were excluded from the review. Prior to data extraction, both reviewers collaboratively developed a comprehensive coding protocol. To ensure reliability, the reviewers tested this protocol on a subset of 39 studies before proceeding with the full data extraction process.

## 3. Results

### 3.1. Study Selection and Characteristics

The study selection process is illustrated in Figure 1. A total of 317 potentially relevant studies were identified through database searches (n = 300) and Google Scholar (n = 17). After removing three duplicate records, 314 unique studies were screened based on their titles and abstracts. Of these, 278 studies were excluded due to irrelevance or failure to meet inclusion criteria. The full texts of 39 articles were assessed for eligibility, resulting in 9 articles being included in this scoping review. Of these, eight studies were retrieved from databases and one from Google Scholar. The included studies were published between 2015 and 2023 and represented a range of geographic regions, including Asia (e.g., Hong Kong, Mainland China, and Taiwan), North America (e.g., Canada), and Europe (e.g., the Netherlands). Sample sizes varied widely, from small pilot studies involving 15 participants to large-scale analyses with over 29,000 participants.

**Figure 1 healthcare-13-02140-f001:**
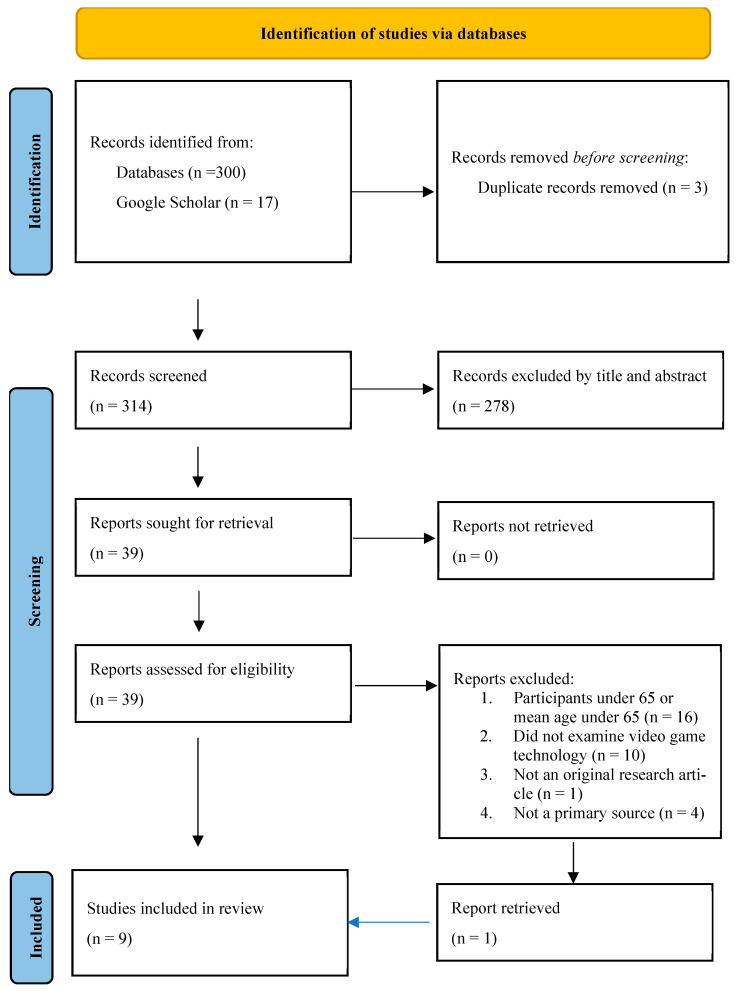
PRISMA-ScR flow diagram for new systematic reviews, which included searches of databases and registers only. Source: Page MJ, et al., 2021 [48].

### 3.2. Intervention Types

The included studies employed a variety of interventions to address loneliness among older adults, categorized as exergames, social and digital games, mobile applications, and general internet use. Each intervention type targeted specific aspects of loneliness and social connection, reflecting diverse methodologies and participant needs.

Exergames were the most common intervention type, appearing in four of the nine studies. These games, which combine physical activity with gaming, were implemented using platforms such as Nintendo Wii and Kinect. Multiplayer exergames demonstrated the most substantial impact on loneliness. For instance, Schell et al. [49] evaluated the effects of an eight-week Wii Bowling program where participants played in small teams. This study reported a significant reduction in loneliness (*p* < 0.001), with participants emphasizing the importance of camaraderie fostered through team-based gameplay. Similarly, Zhu et al. [50] found that older adults who participated in multiplayer Kinect exergames experienced greater social benefits than those in single-player formats, underscoring the role of collaboration in reducing isolation. However, not all exergame interventions yielded positive results for loneliness. Zhu et al. [50] also observed that, while cognitive frailty was mitigated through these games, loneliness outcomes remained insignificant, particularly in groups where the social element of gaming was absent.

Social and digital games also featured prominently in the reviewed studies. These games emphasized multiplayer and collaborative interaction, making them effective tools for fostering social connectedness. Díaz et al. [51] explored the potential of intergenerational gaming through the IRAGE platform, which facilitated collaborative storytelling between older adults and children. Participants in this study reported significant reductions in loneliness, with many highlighting the meaningful connections formed through co-creating narratives. Janssen et al. [52] investigated a chat-based digital game called PhotoSnake, designed to encourage playful interaction through photo-sharing challenges. Although participants initially found the game engaging, long-term engagement was limited due to the perceived superficiality of the interactions. This finding underscored the need for digital games to foster deeper connections to achieve sustained reductions in loneliness.

Mobile applications emerged as another intervention type, with Yang et al. [53] examining their role in mitigating loneliness during the COVID-19 pandemic. This study, conducted in Hong Kong, revealed that frequent use of communication apps such as WhatsApp was strongly associated with reductions in emotional loneliness, particularly among less-educated older adults. Entertainment apps, including YouTube, were also linked to positive outcomes, though their impact was less pronounced than that of communication-focused apps. Despite these benefits, this study highlighted barriers related to technological proficiency, with participants expressing challenges in navigating complex app interfaces. This underscores the importance of designing user-friendly applications tailored to older adults’ needs.

General internet use and gaming were examined in two studies, focusing on their broader impact on loneliness. Fan and Yang [54] analyzed data from the China Health and Retirement Longitudinal Study (CHARLS), which included over 29,000 participants from rural areas. This study found that online gaming had a stronger positive effect on reducing loneliness compared to passive activities such as video streaming.

Across these intervention types, common themes emerged regarding their effectiveness in addressing loneliness. Interventions that emphasized social interaction, such as multiplayer exergames and intergenerational gaming, consistently demonstrated greater reductions in loneliness than single-player or passive formats. Cognitive integration was another key feature, particularly in exergames, which often targeted both physical and mental well-being. However, several studies also identified barriers to sustained engagement, particularly in digital and mobile interventions, where user experience and interface design played a critical role. These findings underscore the importance of tailoring interventions to the unique needs of older adults, leveraging technology to enhance social connection while addressing accessibility challenges.

### 3.3. Moderators and Mediators

The effectiveness of gaming interventions in reducing loneliness among older adults was influenced by various moderators and mediators, including age, gender, and game design. These factors shaped the extent to which participants benefited from the interventions and highlighted the importance of tailoring interventions to individual needs and preferences.

Age was a consistent moderator across several studies. Older participants, particularly those aged 75 and above, reported greater reductions in loneliness compared to younger cohorts (aged 60–74). Schell et al. [49] found that this age-related difference could be attributed to higher baseline levels of loneliness among older participants, which allowed for greater observable changes following interventions. Similarly, Zhu et al. [50] noted that older adults with cognitive frailty experienced significant improvements in cognitive function, but loneliness outcomes were more pronounced in those who engaged in multiplayer formats.

Gender differences also played a significant role in shaping outcomes. Women were more likely to report improvements in social connectedness following multiplayer or socially oriented interventions. For example, Díaz et al. [51] observed that female participants were particularly receptive to the intergenerational gaming sessions, often describing the interactions as emotionally meaningful and fulfilling. Conversely, Janssen et al. [52] highlighted that male participants were less engaged in chat-based games like PhotoSnake, potentially due to the game’s perceived lack of depth or competitiveness.

Game design emerged as a critical mediator, particularly the format and social components of the intervention. Multiplayer and collaborative formats consistently outperformed single-player or passive activities in reducing loneliness. For instance, Díaz et al. [51] emphasized the importance of shared activities, such as storytelling, in fostering meaningful connections between generations. Similarly, Schell et al. [49] demonstrated that team-based Wii Bowling sessions created opportunities for participants to bond over shared goals, significantly enhancing their sense of social belonging. In contrast, Zhu et al. [50] reported that single-player exergames, while effective in improving cognitive outcomes, had a limited impact on loneliness due to the absence of interactive elements.

Digital Literacy also mediated the effectiveness of interventions, particularly those involving mobile applications. Yang et al. [53] found that older adults with higher technological proficiency were better able to engage with communication apps like WhatsApp, leading to greater reductions in emotional loneliness. Conversely, participants with limited digital literacy faced barriers that hindered sustained use and reduced the overall effectiveness of the intervention. Social Interaction was a key mediating factor in the success of gaming interventions. Studies such as Díaz et al. [51] and Janssen et al. [52] highlighted the importance of fostering meaningful interactions between participants. Games that encouraged collaboration, such as intergenerational storytelling or multiplayer exergames, were more effective in reducing loneliness than those that focused solely on individual achievements. For example, Janssen et al. [52] noted that while PhotoSnake initially engaged participants, its limited focus on fostering deep social connections ultimately hindered its long-term impact.

### 3.4. Outcomes

The studies reviewed provided a comprehensive understanding of the impact of gaming interventions on loneliness and related outcomes among older adults. While the primary focus was on loneliness reduction, several secondary outcomes, such as cognitive improvements, emotional well-being, and social connectedness, were also reported.

A significant majority of the included studies reported reductions in loneliness as a primary outcome. Multiplayer interventions consistently demonstrated the strongest effects. For example, Schell et al. [49] found that older adults who participated in team-based Wii Bowling sessions showed a significant decline in loneliness levels, attributed to the collaborative and social nature of the activity. Similarly, Díaz et al. [51] observed that intergenerational gaming sessions, facilitated through the IRAGE platform, led to substantial reductions in feelings of isolation. Participants highlighted the importance of the shared storytelling experience, which fostered meaningful connections across generations. These findings emphasize the central role of social interaction in addressing loneliness, with interventions designed to encourage collaboration yielding the most consistent benefits.

However, not all interventions were equally effective in reducing loneliness. Zhu et al. [50] evaluated the impact of exergaming on loneliness among older adults with cognitive frailty. While the intervention significantly improved cognitive functioning, its effect on loneliness was negligible. This study attributed this to the lack of a meaningful social component in the game design. Janssen et al. [52] similarly highlighted the limitations of digital social games in fostering deeper social connections. Although the PhotoSnake game initially engaged participants, its perceived superficiality and lack of personal interaction limited its long-term impact on loneliness. These findings suggest that interventions lacking robust social elements may struggle to achieve meaningful reductions in loneliness, even if they succeed in other domains.

Several studies explored secondary outcomes, particularly improvements in cognitive and emotional well-being. Zhu et al. [50] reported significant enhancements in cognitive function among participants engaged in exergames, as measured using the Montreal Cognitive Assessment (MoCA). This study demonstrated that regular gameplay could help mitigate cognitive decline, although its effect on loneliness remained limited. Díaz et al. [51] also noted improvements in emotional well-being, with participants reporting greater feelings of purpose and self-worth following intergenerational gaming sessions. These findings highlight the potential for gaming interventions to provide broader benefits beyond loneliness reduction, particularly when they incorporate elements of cognitive and emotional engagement.

Engagement and sustainability were recurring challenges across several interventions. Janssen et al. [52] observed a decline in user engagement over time in the PhotoSnake intervention, with participants citing a lack of variety in the gameplay as a key barrier. Similarly, Yang et al. [53] found that while mobile applications such as WhatsApp effectively reduced emotional loneliness during the COVID-19 pandemic, their impact was moderated by participants’ digital literacy. Older adults with limited technological proficiency reported difficulties navigating app interfaces, which hindered sustained use. These findings underscore the importance of designing interventions that are not only engaging but also accessible and user-friendly for older populations (Table 1).

## 4. Discussion

In this review, we examined the current state of research on the impact of digital gaming on loneliness in the older adult population. We focused on three main research questions: (1) What types of digital games are being used to help address loneliness in older adults? (2) How are studies measuring the effects of these digital games on loneliness in older adults? and (3) What outcomes have been reported from these studies?

### 4.1. Types of Digital Games and Their Effectiveness

One important finding from our review is that digital games that emphasized social interaction, such as multiplayer exergames and intergenerational gaming, consistently demonstrated greater reductions in loneliness compared to single-player or passive formats. This suggests that when designing digital games to combat loneliness, incorporating real-time social interaction with others may be a crucial factor to consider. However, the mechanisms through which multiplayer games enhance social connectedness remain underexplored. Future research should investigate whether structured collaboration, team-based engagement, or competitive elements play a greater role in fostering social bonds.

Another finding from our review is that cognitive integration was another key feature in digital games that impacted loneliness, particularly in exergames. This implies that more integrated or complex exergames may be more impactful, and that targeting both physical and mental well-being may have a more significant impact on alleviating loneliness. The finding that incorporating multiple dimensions of physical, mental, and social activity as components of health in older adults is well supported by previous theoretical frameworks regarding successful aging [58,59,60]. However, the disproportionate focus on exergames raises concerns regarding generalizability. In fact, in a recent survey, overall, 13.9% (age-adjusted) of adults aged 65 and older in the U.S. met federal physical activity guidelines for both aerobic and muscle-strengthening activities [61]. Furthermore, older adults who can engage in the physical activity required by exergames may have a healthier baseline of functioning compared to seniors who are perhaps less physically able or simply older in age. Thus, though there were several studies that found exergaming had a positive impact on loneliness, it should be acknowledged that the samples in these studies may not be representative across the diversity of the older adult population.

In addition, many studies failed to explore whether particular gaming mechanics, such as role-playing, cooperative challenges, or immersive storytelling, contribute to sustained engagement and social cohesion. Digital games that facilitate intergenerational interaction or build on older adults’ personal narratives may provide a more engaging experience while reinforcing meaningful relationships. Future research should evaluate how these features influence sustained engagement and long-term loneliness reduction.

### 4.2. Measurement of Loneliness in Gaming Studies

Several studies in our review measured or reported on moderators and mediators of loneliness. Age and gender were found to be two important moderators. The finding that individuals in older age groups reported less loneliness after gaming than those in younger groups suggests that more attention should be focused on developing more inclusive gaming apps that provide hearing and visual accommodations to better serve this group.

Another finding from our review was that most of the studies used the UCLA Loneliness Scale as their primary measure of loneliness. While this scale is highly regarded and displays high validity and reliability, it is possible that other measures of loneliness might result in more nuanced or different findings.

### 4.3. Interpretation of Outcomes and Contradictory Findings

There is a central role that social interaction plays in addressing loneliness, with interventions designed to encourage collaboration yielding the most consistent benefits. Also, it should be noted that studies on exergaming seemed to represent the largest percentage of positive outcomes, particularly those with multiplayer options. While many studies reported positive effects of gaming interventions, some findings were contradictory, suggesting that the relationship between gaming and loneliness is complex and moderated by multiple factors. For example, while multiplayer exergames and intergenerational gaming showed promising results, some digital interventions, particularly single-player games, had limited impact. The inconsistent findings may stem from differences in study methodologies, intervention durations, or participant characteristics (e.g., prior gaming experience, personality traits, or social support networks). Another factor contributing to contradictory findings is the role of digital literacy. Older adults with higher technological proficiency were more likely to benefit from gaming interventions, while those with lower proficiency faced barriers to engagement. This raises concerns about the accessibility of digital gaming interventions, particularly for older adults with limited experience using technology.

## 5. Limitations

There were several limitations to our review. First, we based our search strategy on studies that specifically included “loneliness OR social isolation OR social exclusion OR lonely OR social interaction”. It is possible that studies focusing on related constructs, such as depression or wellness, which did not come up in our search strategy but did address loneliness or social isolation, were inadvertently excluded. Also, we only included research that was reported in English, which could limit the range of studies that we identified in our review. We also did not include dissertations, as they were not subject to peer review. However, dissertations can offer important insights into where future research is headed, and these were not included in our review.

## 6. Future Directions

The findings from our review point to several areas for future investigation. First, we found that while the studies we examined were from a range of countries, there were proportionally fewer studies conducted in the United States, and virtually none conducted in Africa, India, or Europe. The majority of studies were conducted in East Asian countries. According to the existing literature, most of the research and commercial projects on the use of digital games for older people come from developed and wealthy countries such as the United States, the United Kingdom, Australia, Singapore, South Korea, and Japan [44]. Given this, it would be beneficial for these countries to invest more in researching the ways that digital gaming can help alleviate the health burden of loneliness among older adults. Furthermore, research has found that there are variations in reported levels of loneliness across some European countries [62]. This suggests that there may be cultural differences in the way that loneliness present, and this would also be important to take into consideration in future studies examining the impact of digital gaming on loneliness.

As mentioned in the discussion, the majority of studies included in this review used the UCLA Loneliness Scale as their measure of loneliness. There is some evidence to suggest that other measures, such as the de Jong Gierveld Loneliness Scale, may be more appropriate for use in older adults due to its focus on both social and emotional loneliness [63]. Additionally, in a 2023 systematic review, only one of the studies in the review showed the transcultural validity of the UCLA Loneliness Scale, while four studies examined this in the de Jong Gierveld Scale [64]. Future research might include continued exploration and development of culturally sensitive and valid loneliness measures for older adult populations.

Taking into account the relationship between loneliness and other health outcomes is another way to better understand the impact that loneliness has on overall functioning and quality of life [65]. For example, several studies have found a relationship between loneliness in older adults and living alone or being without a spouse [66].

Finally, several studies in this review examined quite an expansive range of ages; some studies looked at populations aged 50, spanning to above 80 years old. There are numerous important differences in social-emotional, mental, and physical functioning between individuals in their fifties—who may still have children at home and may be working full-time—so those in their eighties, who are likely retired and have households with fewer people in them. 

## 7. Conclusions

As populations worldwide continue to age, leveraging digital technology to enhance social connection and emotional well-being is an increasingly important area of inquiry. The findings from this review highlight both the promise and the limitations of gaming interventions in reducing loneliness among older adults. While multiplayer and cognitively engaging games appear to offer the most benefits, significant gaps remain in our understanding of how to optimize game design, measure intervention effectiveness, and ensure accessibility for all older adults. There are unique aspects of the older adult population to consider in creating technology that can support emotional health and social connection. While the power of digital technology to connect the world is clear, researchers have a great opportunity and a responsibility to better understand how to apply this technology to support the special needs of the aging population.

## Figures and Tables

**Table 1 healthcare-13-02140-t001:** Description of the studies included in this review.

Reference (Year)	Study Characteristics			
	Type of Game	Intervention	Outcome Measures	Findings
Diaz et al. [51] (2024)	IRAGE (Intergenerational Remote Access to Gaming Experiences)	Three “phases” or sessions of an “intergenerational communication experience”	Two quantitative and five qualitative questions asking participants about their gaming experience	Remote inter-generational communication can significantly mitigate isolation among OA.
Fan and Yang [54] (2022)	“online games”	No intervention; survey of internet use	Quantitative survey of frequency of internet use	Playing games had a more positive effect on depression levels than watching videos.
Janssen et al. [52] (2023)	Photosnake, a chat-based gaming app	Three sessions for each group: two 1.5-h workshops two weeks apart and one 1-h focus group two weeks after the second session	Selected items from The Older Persons and Informal Caregiver Survey-Short Form (TOPICS- SF); Six-item De Jong Gierveld Loneliness Scale; Questions about social network size; Focus group questions	In-person contact for engagement was important for in-game interaction. Game design should focus on aiding in players create personal interaction moments.
Li, Then, and Foo [55] (2020)	Wii tennis (single and multiplayer)	Two randomly assigned groups played either single-player or multiple-player Nintendo Wii Tennis exergames for 6 weeks	PHQ-9, BSSS, UCLA Loneliness Scale	Social support was not affected by play mode, but there was a significant relationship between social support, loneliness, and depression
Schell et al. [49] (2016)	Wii bowling	Pre/post questionnaires and interviews for control and experimental group after Wii bowling tournament	Demographic survey; Overall Social Connectedness subscale; UCLA Loneliness Scale; Qualitative interviews	Wii bowling associated with decline in loneliness over 8 weeks.
Yang and Jin [56] (2022)	“single-player games, online games”	Survey re: frequency of use of internet	Core Symptoms of Internet Addiction and Related Problems of Internet Addiction subscales of the Revised Chinese Internet Addiction Scale (CIAS-R); Brief Sensation Seeking Scale-Chinese (BSSS-C); UCLA-3 Loneliness Scale	Loneliness may be a mediator in excessive internet and online gaming use among older adults; real-world leisure activities may be effective in reducing internet and gaming use.
Yang et al. [53] (2022)	14 mobile app types, including games (Candy Crush, Mah jong)	Survey re: frequency of use of apps and 6 loneliness questions	Questionnaire assessing frequency and duration of app use; De Jong Gierveld Loneliness Scale	No significant relationship found between use of gaming apps and reported loneliness by older adults.
Xu et al. [57] (2016)	Exergames	A 2 (pre-test vs. post-test) × 2 (young-old vs. old-old) × 3 (play alone vs. play with elderly vs. play with youths)	UCLA Loneliness Scale	Exergaming may affect loneliness differentially across different types of game play and between different aging cohorts.
Zhu et al. [50] (2023)	Exergame: HappyGoGo (LongGood Meditech, Taipei, Taiwan)	Exergaming intervention consisting of two 40-min sessions being conducted every week over eight weeks	MoCA; Chinese Version of the Loneliness Scale	An 8-week exergaming intervention had significant positive effects on cognitive function but did not positively affect loneliness.

## Data Availability

The original contributions presented in this study are included in the article. Further inquiries can be directed to the corresponding author.

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
