# Peer review of "The Role of Digital Gaming in Addressing Loneliness Among Older Adults: A Scoping Review"

_healthcare, 2025, doi:10.3390/healthcare13172140_

Round 1
Reviewer 1 Report
Comments and Suggestions for Authors
The manuscript deals with the role of digital gaming in addressing loneliness among older adults. Loneliness is one of the most common issues among older adults, especially those living alone, and therefore the discussion on addressing loneliness is both relevant and important.
The literature review is comprehensive, and the aim of the article is clearly and specifically defined.
The methodology and the search strategy across various databases are clearly described.
A few suggestions for improvement:
The introduction lacks a theoretical framework that addresses key theories in the field, such as the contribution of active aging, the importance of social interaction, or alternatively, theories related to resource depletion and the tendency to optimize time use based on the resources available. These theories could be relevant to the topic discussed in the manuscript.
In some parts of the text, the writing is not sufficiently academic. In addition, there is repetition of certain sentences. For example, the statement that loneliness is linked to depression appears multiple times. Similarly, the arguments regarding the advantages of gaming are repeated several times. In this context, it would also be beneficial to discuss the disadvantages of relying solely on virtual communication in the absence of real human interaction, and the risks of it, especially for older adults.
Comments on the Quality of English Language
In some parts of the text, the writing is not sufficiently academic. In addition, there is repetition of certain sentences. For example, the statement that loneliness is linked to depression appears multiple times. Similarly, the arguments regarding the advantages of gaming are repeated several times. In this context, it would also be beneficial to discuss the disadvantages of relying solely on virtual communication in the absence of real human interaction, and the risks of it, especially for older adults.
Author Response
Comment 1:
A few suggestions for improvement:
The introduction lacks a theoretical framework that addresses key theories in the field, such as the contribution of active aging, the importance of social interaction, or alternatively, theories related to resource depletion and the tendency to optimize time use based on the resources available. These theories could be relevant to the topic discussed in the manuscript.
Response: Thank you for pointing this out. We have added more of a theoretical framework to contextualize our study on: page 2, paragraph 3, lines 62-71; page 2, paragraph 4, lines 78-79, lines 83-85; page 9, paragraph 3, lines 371-373.
Comment 2:
In some parts of the text, the writing is not sufficiently academic.
Response: Thank you for your comment. We would like to clarify that we intentionally wrote the article in a more accessible manner. However, we would be happy to adjust the language if you could you please specify which parts are not academic enough.
Comment 3:
In addition, there is repetition of certain sentences. For example, the statement that loneliness is linked to depression appears multiple times.
Response: Thank you for pointing this out. We have deleted a reference to loneliness and depression on page 1, paragraph 2.
Comment 4:
Similarly, the arguments regarding the advantages of gaming are repeated several times. In this context, it would also be beneficial to discuss the disadvantages of relying solely on virtual communication in the absence of real human interaction, and the risks of it, especially for older adults.
Response: Thank you for pointing this out. We have added a section reviewing the potential disadvantages of virtual interactions on page 3, paragraph 1, lines 100-105.
Reviewer 2 Report
Comments and Suggestions for Authors
This scoping review provided a comprehensive understanding of the impact of gaming on loneliness and related outcomes among older adults. Here are some suggestions:
1. Line 104, change "main research" to "main research questions"
2. Use a table to summarize different types of gaming interventions, the measurements used to measure loneliness, compare the benefits and limitations.
Author Response
Comment 1:
Line 104, change "main research" to "main research questions"
Response: Thank you. We have made this change to page 3, paragraph 2, line 115.
2. Use a table to summarize different types of gaming interventions, the measurements used to measure loneliness, compare the benefits and limitations.
Response: Thank you for this suggestion. We have added Table 1 on page 8.
Reviewer 3 Report
Comments and Suggestions for Authors
The authors state that “This scoping review examines the current state of research on the impact of digital gaming on loneliness in the older adult population and was conducted in accordance with the 2018 Preferred Reporting Items for Systematic Reviews and Meta-Analyses extension for Scoping Reviews (PRISMA-ScR) guidelines. A total of 317 potentially relevant studies were identified from database searches and of these, 278 studies were excluded due to failure to meet inclusion criteria. The full text of 39 articles was assessed for eligibility, resulting in 12 articles being included in this scoping review. Some important findings from our study include the central role that social interaction plays in addressing loneliness and targeting both physical and mental well-being may have a more significant impact on alleviating loneliness. We also found that while many studies reported positive effects of gaming interventions, some findings were contradictory, suggesting that the relationship between gaming and loneliness is complex and moderated by multiple factors. Recommendations for future research include expanding investigations to outside of East Asia (where the majority of existing studies were con ducted) to the United States, Africa, India, or Europe.”
This is an interesting paper from a societal and scientifical point of view.
I have the following points to make the paper stronger:
Section 1 Introduction:
Refer and relate to the following meta analysis:
Shah, S. G. S., Nogueras, D., van Woerden, H. C., & Kiparoglou, V. (2021). Evaluation of the effectiveness of digital technology interventions to reduce loneliness in older adults: systematic review and meta-analysis. Journal of medical Internet research, 23(6), e24712.
You could take this paper as a starting point and then narrow down to your specific topic: the impact of DIGITAL GAMING on loneliness in the older adult population.
Define and operationalize EFFECTIVENESS of games.
Section 2.2: Why 2014 as a starting point for your scoping review?
Section 3.3 and 3.4 : Present your results also in a Table.
Focus on studies with sufficient validity and reliability!
Section 6: Also refer to Fokkema, T., Gierveld, J. D. J., & Dykstra, P. A. (2013). Cross-national differences in older adult loneliness. In Loneliness Updated (pp. 200-227). Routledge.
while pointing to the necessity of comparing different countries.
And finally: Check your references. For example: There are references to Zhu et al. but this publication lacks in the reference list.
Author Response
Comment 1:
Section 1 Introduction:
Refer and relate to the following meta analysis:
Shah, S. G. S., Nogueras, D., van Woerden, H. C., & Kiparoglou, V. (2021). Evaluation of the effectiveness of digital technology interventions to reduce loneliness in older adults: systematic review and meta-analysis. Journal of medical Internet research, 23(6), e24712.
You could take this paper as a starting point and then narrow down to your specific topic: the impact of DIGITAL GAMING on loneliness in the older adult population.
Response: Thank you for this comment. We have addressed this comment by referring to this study on page 2, paragraph 4, sentences 72-75.
Comment 2:
Define and operationalize EFFECTIVENESS of games.
Response: Thank you for this comment. We have changed "effectiveness" to "effect" on page 3, paragraph 2, line 117 and on page 9, paragraph 1, line 356.
Comment 3:
Section 2.2: Why 2014 as a starting point for your scoping review?
Response: Thank you for this observation. We have responded to this comment on page 3, paragraph 5, lines 143-145.
Comment 4:
Section 3.3 and 3.4 : Present your results also in a Table.
Response: Thank you for this suggestion. We had added Table 1 to page 8.
Comment 5:
Section 6: Also refer to Fokkema, T., Gierveld, J. D. J., & Dykstra, P. A. (2013). Cross-national differences in older adult loneliness. In Loneliness Updated (pp. 200-227). Routledge. while pointing to the necessity of comparing different countries.
Response: Thank you for your comment. We have incorporated a reference to the article above on page 10, paragraph 3, lines 436-439.
Comment 6:
And finally: Check your references. For example: There are references to Zhu et al. but this publication lacks in the reference list.
Response: Thank you for pointing this out. We have added this article to the reference list and have checked our references again.
Round 2
Reviewer 2 Report
Comments and Suggestions for Authors
The authors provided poin-by-point responses to the suggestions and improved the manuscript as suggested. The error has been fixed. The table has been added. The revision is sufficient.
Reviewer 3 Report
Comments and Suggestions for Authors
The manuscript is OK now!